# US cities increasingly integrate justice into climate planning and create policy tools for climate justice

Claudia V. Diezmartínez [1] ✉ & Anne G. Short Gianotti[1]

Climate change is one of the most important ethical issues of our time. Urban scholars and policymakers now recognise the need to address justice concerns associated with cities' responses to climate change. However, little empirical research has examined whether and how cities have integrated justice into climate mitigation planning. Here, we show that large cities in the US are increasingly attending to justice in their climate action plans and that the recognition of structural and historical injustices is becoming more common. We demonstrate that justice is articulated differently across mitigation sectors, uncover local characteristics that may impact cities' level of engagement with justice, and introduce four policy tools that pioneer cities have developed to operationalise just climate policies on the ground. More attention to justice in policy implementation and evaluation is needed as cities continue to move toward just urban transitions.

Climate change is increasingly understood as intertwined with concerns about justice and equity[1–5]. It is widely known that climate change is disproportionately impacting the most vulnerable populations worldwide, even as many of these groups have contributed the least to global greenhouse gas emissions[5–7]. More recently, the linkages between equity and climate responses, including both actions taken to mitigate and to adapt to climate change, have been recognized. Climate efforts produce benefits and burdens, distribute resources, reorganize space, and impact infrastructure with uneven consequences across communities and populations. Climate action thus has the potential to exacerbate or redress existing social inequities and vulnerabilities[4,7–10].

The nexus of climate action and justice is particularly pronounced in cities. Most of the world's population lives in cities and urban areas generate more than 70% of global $CO_2$ emissions[11]. Cities have been important sites of climate action for more than two decades[12–15] and recent efforts such as the United Nations Race to Zero and the Race to Resilience aim to spur a transition to net-zero cities and catalyse urban climate projects. The actions cities take to reduce emissions and adapt to climate change will produce benefits as well as unintended consequences that are likely to be distributed unevenly within and beyond city boundaries[4,5,10,16].

In light of the growing attention to climate justice at the global, national, and local scales, many city governments, advocates, and scholars have made bold calls for "just urban transitions"[8,17,18] or a "green and just recovery"[19] following COVID-19. Informing and evaluating progress on this agenda requires a deep understanding of urban climate planning and action. While previous research suggests few cities have meaningfully incorporated equity or justice goals into their climate strategies[4,20–26], most analyses of urban climate plans focus primarily on climate adaptation and resilience[18,21,23,24,26–29] or sustainability more broadly[30,31]. The few studies that examine climate mitigation plans have been limited to a relatively small number of cities[20,22,25] or a few specific mitigation sectors[32]. We thus lack a comprehensive picture of how justice concerns have been integrated and institutionalised into urban climate action planning, and this has translated into a paucity of policy guidance on how cities can pursue more just urban transitions[8,28].

Addressing this gap in knowledge is critical. Cities' policies to mitigate greenhouse gas emissions intersect with many aspects of urban life and redistribute resources with direct and indirect consequences for vulnerable populations. There is growing evidence that urban climate actions can lead to disparities in energy access and pricing[9,33,34], inequitable access to clean technologies[9,34,35] and low-

¹Department of Earth and Environment, Boston University, Boston, MA, USA. ✉e-mail: cvdiezm@bu.edu

carbon transportation[36,37], unequally distributed employment opportunities[9,20,38] and green gentrification[36,39]. Urban scholars and decision-makers could therefore benefit from understanding how different cities conceptualize the justice implications of climate mitigation policies and identifying the policy tools that have been developed to address these complex issues in urban areas.

Here, we show that large cities across the US are increasingly incorporating justice into their climate action plans and developing policy tools to integrate justice and equity concerns into their climate mitigation policies, particularly in the last five years. We conduct a content analysis of the most recent climate mitigation plans developed by the 100 largest cities in the US and provide a comprehensive assessment of the degree to which cities are attentive to justice in climate action planning. We find that the recognition of cities' historical patterns of racial segregation, disinvestment, environmental injustice, and exclusion is becoming more common in recent plans, although attention to justice is not equally distributed across mitigation sectors. We highlight local factors that may influence cities' level of engagement with justice in their climate action plans and uncover four concrete policy tools cities are using to implement and evaluate work toward "just urban transitions".

## Results and discussion
### Engagement with justice in urban climate action plans
Fifty-eight of the 100 largest US cities had an approved climate action plan as of June 2021 (Supplementary Table 1). For each of these cities, we conducted a content analysis of their most recent plan to evaluate if and how justice and equity are addressed in their climate mitigation policies. We coded climate plans across six main themes: (1) distributive justice; (2) procedural justice; (3) justice as recognition; (4) justice in climate mitigation sectors; (5) key definitions; and (6) key sections where justice is articulated (Supplementary Table 2).

We found a range of engagement with justice in urban climate action plans (Table 1). Forty cities (69%) are attentive to justice in their climate action plans, either by *aspiring for justice* (20 cities, 34.5%) or by *explicitly planning for justice* (20 cities, 34.5%). The 20 cities that *aspire for justice* articulate justice and/or equity as a goal, vision, guiding principle, or core value of their plan but do not explicitly describe policy actions or systematic strategies to implement or evaluate progress toward just climate mitigation. The 20 cities that are *planning for justice* systematically embed justice into the design of their climate policies by using justice and/or equity as a criterion to select policy interventions and/or by using justice focused policy tools to develop and operationalise climate action policies. Eighteen cities (31%) *do not articulate justice as a core feature of climate action*. These cities do not describe justice or equity as an objective of their plan and lack policy measures explicitly aimed at addressing justice concerns (Supplementary Table 3).

Justice has become a more common feature of climate action plans in recent years. Thirty-one of the 40 plans (78%) that incorporate justice were published between 2017 and 2021 (Fig. 1). Of the 22 plans published before 2017, only 22.7% articulated justice as an aspiration and 18.2% explicitly planned for justice. In contrast, of the 36 plans that were adopted between 2017 and 2021, 41.7% articulated justice as an aspiration and 44.4% explicitly planned for justice.

Using ordinal logistic regression, we confirm that the time of publication of the climate action plans is a significant determinant of cities' level of engagement with justice, even after accounting for cities' sociodemographic, economic, and political characteristics (Table 2). Although previous studies have found limited evidence of clear relationships between city characteristics and their degree of focus on justice in climate mitigation and adaptation planning[20,27], we find that several local factors may increase the likelihood of cities incorporating justice into their climate action plans. First, cities with a higher median household income and cities with higher levels of poverty have increased odds of incorporating justice into their climate plans. This suggests that cities with more economic inequities (i.e., high incomes and high poverty rates) are paying more attention to justice. Similar to Hess and Mckane[32], we find no evidence that higher population diversity positively impacts cities' level of engagement with justice. However, our model supports the finding by Liao et al.[40] that public engagement in climate planning is associated with greater attention to justice. We also find that cities with larger populations are more likely to have higher levels of engagement with justice. This may be due to the higher capacities of large cities to undertake more complex planning efforts[28,40], but it might also reflect broader trends of big cities increasingly creating climate action plans in general[15,17]. Finally, we find that coastal cities have increased odds of engaging with justice, while legacy cities (i.e., post-industrial cities) have decreased odds. This could be explained by cities' differential levels of vulnerability to climate change and governance capacities. Case studies in these different types of geographies could help understand and disentangle the complex dynamics of climate action and justice planning in these contexts.

### Articulations of justice and equity
Cities tend to use the language of "equity", rather than "environmental justice" or "climate justice". We find that when cities provide a definition for these concepts, they generally define "justice" as prioritising historically vulnerable communities and those disproportionately affected by climate change, while "equity" tends to be more broadly defined as ensuring equitable access and distribution of the benefits of climate policies. Cities' articulation of "equity" in lieu of "justice" aligns with previous analyses of climate adaptation plans that found that discourses around the distribution of benefits and burdens of climate efforts dominate over deeper accounts of structural injustice[18,23,24,27,28]. Nonetheless, we also find that 15 cities (26%) recognise and articulate their histories of racial segregation, disinvestment, environmental injustice, and exclusion. For instance, the climate plans of Portland (2015), Dallas (2020), and Washington D.C.

**Table 1 | Cities categorised by their level of engagement with justice in their climate action plan**

| Cities that do not articulate justice as a core feature of climate action | Cities articulating justice as an aspiration | Cities explicitly planning for justice |
|---|---|---|
| Austin, TX | Charlotte, NC | Anchorage, AK |
| Boise City, ID | Chula Vista, CA | Atlanta, GA |
| Chesapeake, VA | Columbus, OH | Baltimore, MD |
| Chicago, IL | Denver, CO | Boston, MA |
| Durham, NC | Detroit, MI | Cincinnati, OH |
| Fremont, CA | Indianapolis, IN | Cleveland, OH |
| Greensboro, NC | Madison, WI | Dallas, TX |
| Kansas City, MO | Milwaukee, WI | Houston, TX |
| Louisville, KY | Newark, NJ | Los Angeles, CA |
| Miami, FL | New Orleans, LA | Memphis, TN |
| Pittsburgh, PA | Norfolk, VA | Minneapolis, MN |
| Raleigh, NC | Oklahoma City, OK | New York, NY |
| Richmond, VA | Orlando, FL | Oakland, CA |
| Riverside, CA | Plano, TX | Philadelphia, PA |
| San Jose, CA | Reno, NV | Portland, OR |
| Santa Ana, CA | Sacramento, CA | San Antonio, TX |
| Stockton, CA | San Francisco, CA | San Diego, CA |
| Winston-Salem, NC | St. Louis, MO | Seattle, WA |
|  | St. Paul, MN | St. Peterburg, FL |
|  | Tampa, FL | Washington, DC |

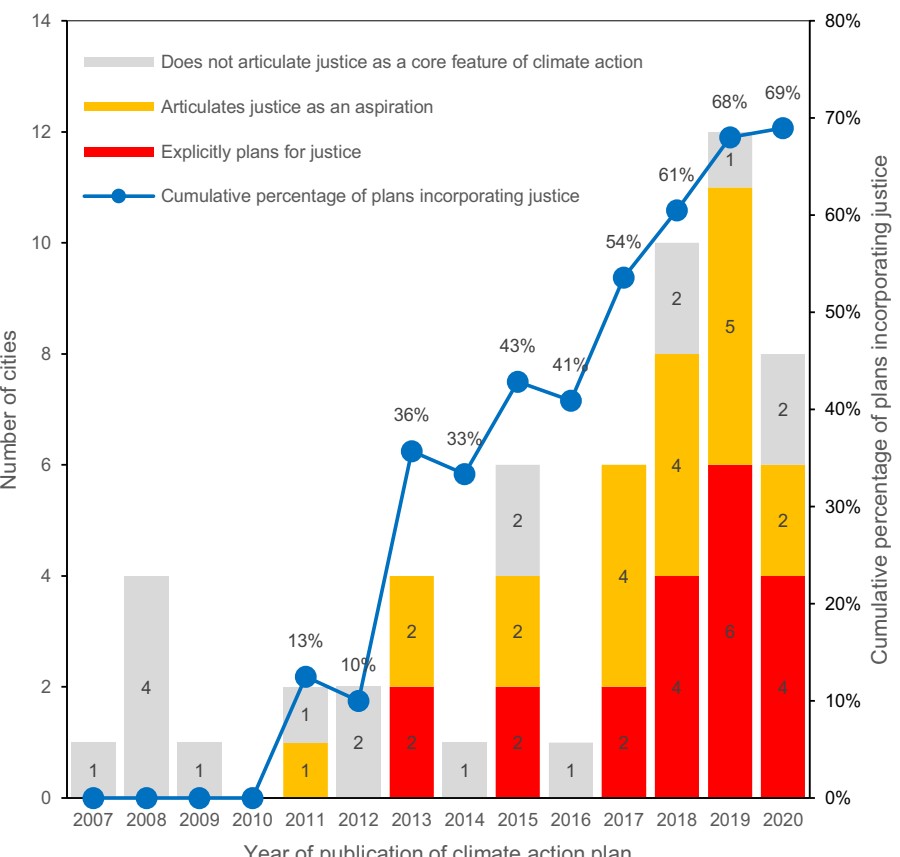

**Fig. 1 | Engagement with climate justice over time.** Number of cities in our sample that adopted or updated a climate action plan between 2007 and 2020. Cities are categorised according to their level of engagement with justice in policy action: *cities that do not articulate justice as a core feature of climate action* (grey), *cities* *articulating justice as an aspiration* (yellow), and *cities explicitly planning for justice* (red) (left axis). The blue line indicates the cumulative percentage of plans incorporating justice in any way (right axis). By 2020, 69% of all plans published between 2007 and 2020 include justice.

## Table 2 | Local factors and climate plan characteristics and their effect on cities' level of engagement with justice in climate action planning

| Variable | Variable Description | Regression Results | | | | |
|---|---|---|---|---|---|---|
| | | Coefficients | Standard Error | Z value | P-value | Odds Ratio |
| After 2017 | Climate plan published after 2017 | 2.3984 | 0.7317 | 3.278 | 0.0010 ** | 11.0053 |
| Population | City population > 500,000 | 1.4528 | 0.6446 | 2.254 | 0.0242 * | 4.2749 |
| Median household income (MHI) | MHI > sample mean ($61,532) | 3.2287 | 1.0585 | 3.050 | 0.0023 ** | 25.2472 |
| Poverty | Percentage of persons in poverty, 2019 | 0.3997 | 0.1176 | 3.399 | 0.0007 *** | 1.4914 |
| People of colour | Percentage of population who did not identify as "White alone, not Hispanic or Latino" in the US Census. This includes African American, Native American, Native Hawaiian and Pacific Islander, Asian, Hispanic or Latino, or two or more races, 2019 | −0.0660 | 0.0245 | −2.691 | 0.0071 ** | 0.9361 |
| Coastal city | City is geographically located by the coast | 2.7406 | 0.8063 | 3.399 | 0.0007 *** | 15.4966 |
| Legacy city | City has been classified as a legacy city | −1.5219 | 0.9222 | −1.650 | 0.0989 | 0.2183 |
| Engagement | City mentions engaging with local community members for the climate plan | 3.6967 | 1.4454 | 2.558 | 0.0105 * | 40.3139 |
| Intercept Category 1 \| 2 | | 9.143 | 2.755 | 3.319 | 0.0009 *** | 9,345.4134 |
| Intercept Category 2 \| 3 | | 11.816 | 2.969 | 3.980 | 0.00006 *** | 135,464.8602 |
| McFadden Pseudo-R² | | | 36.72% | | | |

Dependent variable is an ordinal variable that classifies cities' level of engagement with justice in their climate plans into three categories: Category 1: cities that do not articulate justice as a core feature of their climate plan; Category 2: cities that articulate justice as an aspiration; Category 3: cities that are explicitly planning for justice. Level of significance denoted as follows: *** *p-value* < 0.001; ** *p-value* < 0.01; * *p-value* < 0.05. P-value was calculated through two-sided z-test (*Z* >|z|, α = 0.05).

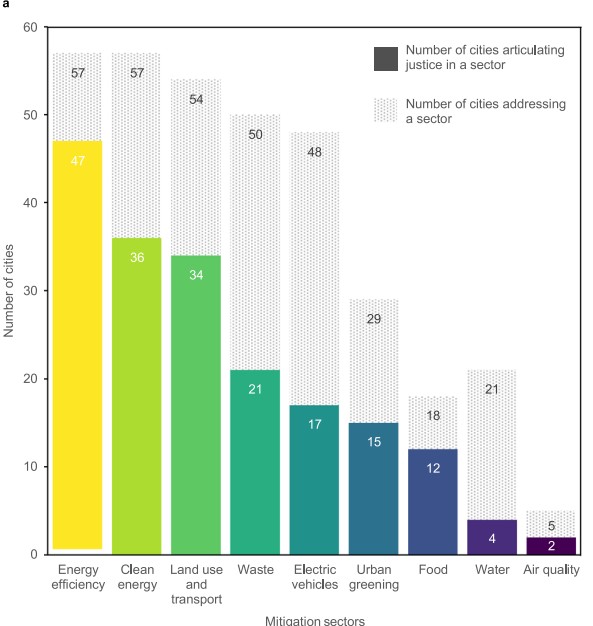

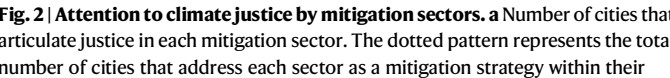

**Fig. 2 | Attention to climate justice by mitigation sectors. a** Number of cities that articulate justice in each mitigation sector. The dotted pattern represents the total number of cities that address each sector as a mitigation strategy within their climate action plan. Solid colours represent cities that articulate justice in each sector. **b** Main themes and policies discussed with respect to justice within each mitigation sector.

(2018) incorporate narratives of their own institutional discriminatory practices and identify the specific neighbourhoods or census tracts that have been historically disadvantaged within their boundaries. This attention to the history of structural injustice is recent, with 12 of the 15 plans (80%) that articulate narratives of structural injustice published in or after 2018.

Cities that recognise historical and current injustices are primarily focused on racial and income inequalities, with less consistent attention to vulnerabilities and injustices associated with gender, age, or disability. This emphasis on racial and economic justice has also been identified in climate adaptation plans[18,21,28,41], perhaps reflecting US cities' long history of racial discrimination, segregation, and income inequalities, as well as the rise of grassroots movements demanding city governments to address these structural issues[18,28,42]. The deficit of narratives connecting gender and disability with climate mitigation is noteworthy, but it is not unique to cities. Research has found that, from local to international spheres, few mitigation policies and regulations refer to gender, suggesting that the role of women is better recognised in adaptation than in mitigation[6,43]. Similarly, scholars have identified a dearth of policy actions that are inclusive of people with disabilities in both climate mitigation and adaptation[44–46].

While most plans analysed here were published before the outset of COVID-19, the City of Oakland's climate plan (2020) incorporates a narrative of how the pandemic has served to highlight the pervasive inequalities and disproportionate burdens experienced by "people of colour, small business owners, and income-insecure workers", and to further underscore the need for climate action "underpinned by climate equity and environmental justice". We can conjecture that new climate plans developed amid or after the COVID-19 pandemic will articulate similar narratives and include deeper accounts of structural injustice, particularly with respect to racial and economic inequities.

## Attention to justice across mitigation sectors

We identified nine major mitigation sectors that US cities have included in their climate action plans: (1) energy efficiency ($n = 57$); (2) clean energy ($n = 57$); (3) land use and transport ($n = 54$); (4) waste ($n = 50$); (5) electric vehicles ($n = 48$); (6) urban greening ($n = 29$); (7) food ($n = 18$); (8) water ($n = 21$); and (9) air quality ($n = 5$). While equity concerns intersect multiple sectors[32,47], we find that cities' attention to justice is not distributed uniformly across policy areas (Fig. 2a). The most common sectors where cities connect mitigation to justice concerns are energy efficiency (47 out 57 plans addressing this sector incorporate justice), clean energy (36 out of 57), and land use and transport (34 out of 54). In contrast, less than half of the cities we analysed link justice to policies related to waste (21 out of 50), electric vehicles (17 out of 48), water (4 out of 21), and air quality (2 out of 5). Although relatively few cities address urban greening and food as part of their mitigation strategies, more than half of these cities connect these policy areas to justice (15 out of 29 and 12 out of 18, respectively).

Figure 2b presents the main themes and policies that cities articulate with respect to justice for each mitigation sector. Cities primarily focus on addressing the direct justice impacts of climate action policies (e.g., energy burdens, access to technologies and services, etc.). Explicit attention to indirect impacts such as displacement and gentrification have received less attention overall ($n = 10$) and these discourses are most often connected to energy efficiency and land use and transport interventions.

Several cities have also developed programs directed at targeted workforce development and outreach efforts. Fourteen cities (24%) include green jobs training programs for vulnerable populations such as people of colour, low-income residents, individuals with barriers to employment, women, youth, veterans, and workers affected by the energy transition. For example, the City of Madison's (2018) Green-Power Program hires under- and unemployed individuals and provides them with training for solar installation jobs. Eighteen cities (31%) also plan to undertake targeted outreach efforts aimed at informing historically vulnerable populations about available climate programs. For instance, the City of Dallas' plan (2020) includes the development of special engagement programs to reach low-income residents, the senior community, and non-native English speakers, and provide them with information about new weatherization programs. Through this "focused engagement", the city expects to address common barriers to program participation and ensure that the benefits of weatherization reach those who need them the most.

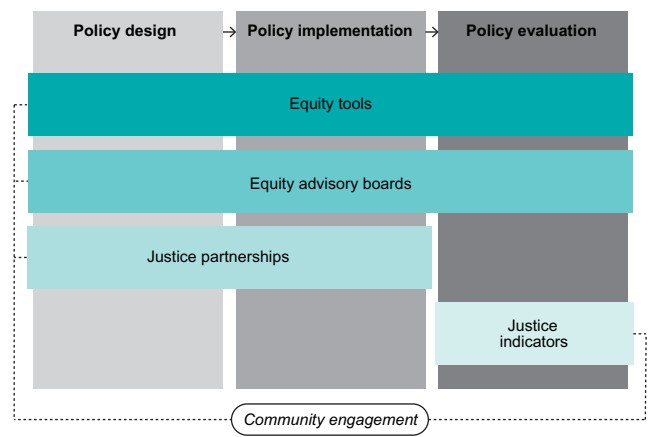

**Fig. 3 | Policy tools and strategies to develop just climate policies.** *Equity tools* and *equity advisory boards* can be implemented throughout the policy process. *Justice partnerships* are mainly focused on policy design and implementation. *Justice indicators* are used during policy evaluation. All policy tools and strategies may involve community engagement.

## Operationalising just climate policies

Several cities have already developed policy tools to implement and evaluate just climate policies. This finding is important, as scholars and practitioners involved in both climate adaptation and mitigation have repeatedly called out the lack of concrete tools and metrics to operationalise just climate policies on the ground[5,8,17,20,28,32,48]. We categorise the policy tools we identified into four types: *justice partnerships, equity advisory boards, equity tools*, and *justice indicators*. Cities may use these instruments at one or more stages of the policymaking process, and they often involve the engagement of multiple community actors (Fig. 3).

Seventeen of the 20 cities that are explicitly planning for justice describe leveraging *justice partnerships* to implement just climate policies. Community partnerships are a common strategy to operationalise climate policies overall. In fact, 40 cities in our sample (69%) mention the need to cooperate with local actors to reach their climate mitigation goals. In the context of justice, community partnerships are seen not only as practical necessity, but also as a tool to promote participation among historically underrepresented communities and to provide legitimacy to policies and programs. Justice partnerships are thus mainly focused on engaging with vulnerable groups, either directly or through environmental and social justice advocacy groups that represent them.

A second strategy to incorporate justice concerns into the operationalisation of climate policies is to create *equity advisory boards* (Table 3). These are groups of community members convened by city governments to facilitate the planning, implementation, and/or evaluation of just climate policies. Equity advisory boards are usually organised into one or more committees, subcommittees, or working groups and are granted varying levels of involvement throughout the policy process. In general, however, these boards are tasked with representing and engaging vulnerable populations, proposing justice centred policy objectives and actions, and reviewing policies and programs to ensure they are aligned with justice goals. In some cases, equity advisory boards are also responsible for developing equity tools themselves. Although the members of these boards are not explicitly listed in all climate action plans, we found that these groups are commonly comprised by residents, advocates, academics, representatives from the private sector, and government officials.

Six of the cities that are explicitly planning for justice have also developed or are in the process of developing an *equity tool* (Table 4). We define equity tools as decision-making frameworks that guide city

governments to recognise and systematically incorporate justice and equity concerns throughout the policy process. Even though the scope of these instruments varies across cities, equity tools usually consist of a set of guiding questions or checklists that provide the basis for creating justice centred policies, identifying and engaging local community actors, developing implementation strategies, and/or analysing the justice and equity impacts of programs. Equity tools are one of the most novel strategies primarily geared toward the operationalisation of just climate policies. A recent example is the city of San Antonio's "Climate Equity Screening Tool", which includes over 30 guiding questions designed to identify the benefits and unintended consequences that policies may produce for local vulnerable groups. This tool will be used by key community actors to evaluate each strategy outlined in the climate action plan prior to implementation.

Finally, eight cities have created or are planning to create *justice indicators*. These are comprehensive metrics to monitor and evaluate the justice and equity impacts of climate plans and policies (Supplementary Table 4). Unlike equity tools that provide broad guidelines to assess the consequences of climate programs, justice indicators enable cities to quantitatively measure the benefits and costs that climate policies bring to vulnerable populations and to track their progress toward their justice goals. For instance, the city of San Diego developed a "Climate Equity Index" to identify vulnerable communities across the city and measure the equity impacts of its climate policies over time. The index integrates over 30 standardised indicators covering multiple environmental, housing, mobility, socioeconomic, and health factors that are used to calculate a climate equity score for each of the census tracks within the city.

## Moving toward just urban transitions

Our systematic analysis of climate action plans reveals that a transition toward urban climate justice governance is emerging in the US. This research complements insights from recent studies focused on climate adaptation planning and builds a comprehensive and nuanced picture of urban climate justice efforts across large cities in the US. Over the past two decades, cities engaged in climate planning have not only paid attention to justice with respect to climate adaptation. Justice and equity concerns have also played an increasingly important role in the design of urban climate action plans, progressively pushing cities to articulate more just and inclusive mitigation actions and to develop policy tools to implement and evaluate climate justice efforts.

While the increasing attention to justice is promising, we highlight three important caveats in this optimistic result. First, 42 out the 100 largest US cities have yet to adopt a city-wide climate action plan. Although urban climate plans have often been found to lack implementation guidance[48] and planning practice itself has led to mixed results in advancing climate action in the past[15,28,49], the development of climate plans is still considered a critical step to systematise urban responses to climate change, provide engagement opportunities to local actors, and legitimise climate policies[25,28,32,48]. Previous research shows that when municipalities report equity as a priority or goal in a formal planning document, they are more likely to adopt more actions related to social equity[40]. Furthermore, climate plans provide a unique avenue to institutionalise justice-focused goals that can mobilise multiple community actors towards this collective purpose[50]. Our findings regarding the increasing attention to climate justice and the recognition of structural injustice in cities suggest that urban policymakers and activists should regard local climate plans as a key tool to advance just urban transitions in their communities.

The second caveat emerging from our analysis is that there is a need for more comprehensive approaches to justice across and beyond climate mitigation sectors. Local climate actions plans are commonly organised by sector-specific chapters that reflect city government's own divisions across departments[32]. Our results indicate that this practice has translated into sector-specific articulations

**Table 3 | Overview of equity advisory boards**

| City | Name of body | Policy stage | Members identified in climate plan | Main tasks and responsibilities |
|---|---|---|---|---|
| Anchorage, AK | (a) Steering Committee (b) Advisory Committee (c) Working Group | Design, implementation, and evaluation | Community members, advocacy groups, academics, private sector, government officials. | (a) Community engagement and design of Equity Implementation Guide. (b) Review of plan drafts, equity-centred policy advice, implementation assistance. (c) Crafting equity-centred policy objectives and actions. |
| Atlanta, GA | Advisory Group | Design and implementation | Community members, advocacy groups, private sector. | Crating policy goals, policy implementation advice, policy analysis and review. |
| Baltimore, MD | Sustainability Ambassadors | Design | Community members | Community representation and engagement. |
| Boston, MA | Community Working Group | Design | Community members, advocacy groups, academics, private sector, government officials. | Policy design and advice. |
| Charlotte, NC* | Workforce Development Working Group | Design | Advocacy groups, academics, private sector, government officials. | Ensuring equitable access and distribution of jobs. |
| Cleveland, OH | Equity and Engagement Subcommittee | Design and implementation | Advocacy groups, academics, private sector, government officials. | Design of Racial Equity Tool. |
| Dallas, TX | Environment and Sustainability Committee | Design and implementation | Government officials | Policy design, policy implementation guidance and assistance. |
| Los Angeles, CA | Climate Emergency Commission | Implementation | Community members, private sector, government officials | Community representation and engagement during implementation. |
| Minneapolis, MN | (a) Environmental Justice Working Group (b) Community Environmental Advisory Commission | Design and evaluation | Community members, advocacy groups, academics, government officials | (a) Community representation, policy design and advice, policy review. (b) Plan revision. |
| Oakland, CA | (a) Ad hoc Advisory Committee (b) Equity Facilitator (c) Neighbourhood Leadership Cohort | Design and implementation | Community members, advocacy groups | (a) Review of plan drafts, policy advise. (b) Community engagement, policy review, and design of the Racial equity impact assessment and implementation guide. (c) Community engagement. |
| Portland, OR | Equity Working Group | Design and implementation | Advocacy groups | Policy design and advice, design of Equity implementation guide. |
| San Diego, CA | Equity Stakeholder Working Group | Design and evaluation | Advocacy groups | Policy design and advice, design of the Climate Equity Index. |
| San Antonio, TX | Climate Equity Advisory Committee | Implementation | Community members, advocacy groups | Community representation, implementation guidance and assistance. |
| St, Louis, MO | Climate Action Planning Equity Advisory Committee | Evaluation | Not stated in the plan | Measurement of policy impacts. |
| St. Paul, MN | Advisory Group | Implementation | Community members | Policy implementation guidance and assistance. |
| Washington, DC | Equity Advisory Group | Design and implementation | Community members | Community representation and engagement, policy design, implementation guidance and assistance. |

*This equity body is only focused on workforce development policies.

**Table 4 | Overview of equity tools**

| City | Name of tool | Policy stage | Main guiding themes | Authors and sources |
|---|---|---|---|---|
| Anchorage, AK | Equity Implementation Guide | Implementation and evaluation | Equity analysis, identification of community members, community engagement, evaluation. | Will be developed by the plan's Steering Committee and adapted from Portland's equity framework. |
| Baltimore, MD | EquityLens | Design, implementation, and evaluation | Community engagement, data gathering, accessibility, capacity-building, priorities of vulnerable populations, disproportionate impacts, economic opportunity, displacement, accountability. | Adapted from the Government Alliance for Race and Equity's "Equity Toolkit" and Portland's "Climate Equity Considerations". |
| Cleveland, OH | Racial Equity Tool | Design and implementation | Language, accountability and data, disproportional impacts, economic opportunity, neighbourhood engagement. | Developed by the plan's Equity and Engagement Subcommittee and adapted from the Government Alliance for Race and Equity and Portland's "Climate Equity Considerations". |
| Oakland, CA | (a) Racial Equity Impact Assessment and Implementation Guide (b) Racial Equity Implementation Guide | Design, implementation, and evaluation | (a) Equitable governance, community engagement, equitable investments, community resilience. (b) Racial equity outcomes, community engagement, data gathering, equity gaps, accountability. | (a) Developed by the plan's Equity Facilitator and adapted from the California Office of Planning and Research's "Resiliency Guidebook Equity Checklist", the NAACP's "Our Communities, Our Power"; the Movement Strategy Center's "Spectrum of Community Engagement to Ownership"; and material from the City of Oakland Department of Race and Equity. (b) Developed by the city of Oakland Department of Race and Equity. |
| Portland, OR | Equity Implementation Guide | Implementation | Data gathering, accessibility, capacity-building, effective partnerships, equitable distribution of costs and benefits, community wealth building. | Developed by the plan's Equity Working Group. |
| San Antonio, TX | Climate Equity Screening Tool | Implementation and evaluation | Access and accessibility, affordability, cultural preservation, health, safety and security. | Not stated in the plan. |

of justice, an uneven attention to justice and equity across mitigation sectors, and little emphasis on the indirect impacts of policies (Fig. 2). The articulation of sector-specific justice concerns is also present in climate adaptation planning[23,28]. Previous analyses of climate adaptation plans have found that cities commonly articulate justice within the context of public health, affordable housing, transit, green infrastructure, and economic opportunities[23,27]. This aligns with cities' attention to the burdens that energy efficiency, clean energy, transportation, and urban greening policies may impose on low-income households, as well as their focus on the equitable distribution of employment opportunities created by climate mitigation. However, we find that cities devote less consistent attention to public health in climate mitigation plans, which may explain the relatively few references to justice issues related to the food, water, and air quality sectors. These sectoral approaches across climate mitigation and adaptation plans are not always adequate to address the justice implications of climate change and climate policy because issues may arise at the intersection of two or more sectors or due to aggregation of multiple climate interventions[17,28,32,47]. For example, the combination of low-carbon and adaptation policies such as urban greening, transit-oriented developments, and energy-efficient housing, may cause the displacement of low-income residents out of improved neighbourhoods[17,36]. Just urban transitions require shifting away from narrow sector-by-sector approaches and pursuing systemic efforts to transform local economies and urban life itself[17,47]. This calls for urban decisionmakers and scholars to look beyond the direct consequences of specific types of policies and address the broader, cross-sectoral implications of climate action. Investigating why cities devote unequal attention to justice across sectors and the implications of these sectoral differences are important open questions for future research.

A final caveat is that most climate plans in our sample have not yet articulated specific strategies to operationalise just climate policies on the ground. Moving towards just urban transitions entails the development and implementation of tools that can guide urban decisionmakers on how to allocate climate efforts and resources, how to recognise who should be prioritised, who needs to be included and informed about climate efforts, and what trade-offs are necessary to build a just low-carbon society[8,18,23]. Our analysis identified a group of pioneer cities and four concrete implementation tools (i.e., justice partnerships, equity advisory boards, equity tools, and justice indicators) that can serve as models for other cities involved in climate action planning. Because most climate plans and policy tools examined here have been developed only in the past few years, our analysis cannot assess whether and how these tools have been successful at addressing historical and structural injustices, engaging and empowering vulnerable populations, and ultimately enabling socially just outcomes. However, our findings provide a baseline to inform and guide future research focused on just implementation efforts. Case studies in cities such as Oakland, Cleveland, Baltimore, or San Antonio, where just implementation and evaluation tools are being developed, can help address these open questions.

At the same time that cities have evolved into essential sites for global climate policy[8,13,51,52], climate governance itself has become a strategic priority of urban politics[17,53]. As questions of justice and equity in the city rise on the agenda, we can expect that climate justice will also become a fundamental component of urban governance over the next decade[17]. New opportunities arise as the COVID-19 pandemic and recent social movements such as Black Lives Matter increase the salience of systemic injustices and reignite collective calls for justice and social transformation[51]. At this critical time, this research can help urban decisionmakers and other key actors in cities to identify how climate justice can be embedded within local climate action efforts, recognise potential benchmarks and learning opportunities from other cities, and reflect upon the

ways in which local policies may or may not be aligned to pursue just urban transitions.

Our study presents a comprehensive picture of how large cities in the US have integrated justice into climate mitigation planning and provides an important step towards understanding how new policy tools can support the implementation of justice focused urban climate policies. As urban climate justice becomes more prevalent in the US and globally, scholars and urban decisionmakers need to ask new questions about climate governance and identify the best pathways and policy tools that facilitate the implementation and evaluation of just climate policies. Understanding the emerging dynamics of climate justice governance and analysing how innovative policy instruments such as justice partnerships, equity bodies, equity tools, and justice indicators operate on the ground are crucial next steps to support and inform future efforts towards just urban transitions.

## Methods

To examine the emergence of climate justice in urban climate mitigation planning, we analysed local climate action plans adopted by the 100 largest cities in the US. The list of cities included in our study was defined according to the US Census Bureau 2019 population estimates. We focus on large cities because (a) these urban areas are more likely to have more diverse populations that experience relatively pronounced poverty and income disparities[20,27] and (b) their governments are more likely to have more resources and capacities to undergo complex climate planning processes that incorporate justice and equity[28,54]. Moreover, focusing on large cities enables us to compare our findings across previous studies, most of which examine climate planning in large cities[18,20,21,27,28,32].

### Sampling of climate action plans

We built our sample by collecting the most recent climate action plan available for each of the 100 largest cities in the US. We define climate action plan as any formal local planning document adopted by a city government that explicitly addresses multiple sectors of climate mitigation. This definition includes climate plans exclusively focused on mitigation, climate plans integrating mitigation and adaptation or resilience, as well as sustainability and energy plans with chapters or sections explicitly dedicated to climate mitigation. We excluded city plans that are only focused on climate adaptation or resilience, plans that are written by state or regional entities, and plans that are written by local entities (e.g., local non-profits, universities) but not formally adopted by city governments. Our definition of climate action plan enabled us to capture a comprehensive and nuanced picture of cities' discursive representations of climate justice with respect to climate mitigation, while also maintaining a relatively consistent and comparable sample.

We collected plans through targeted internet searches in Google (e.g., "city name" + "climate action plan"), city government websites, and the Local Government Climate and Energy Goals database developed by the American Council for an Energy-Efficient Economy. For each city, we selected the most recently adopted climate action plan that fit our definition as of June 2021. Several cities in our sample had published multiple plans over the past decade. In cases where the most recent plan updated or superseded earlier plans, we reviewed only the most recent plan. However, in cases where the most recent plan complemented an earlier plan, we reviewed both the most recent and previous versions of the plan. In total, we found that 58 out the 100 largest US city had an eligible climate action plan to include in our analysis (Supplementary Table 1).

### Coding protocol and procedures

We coded the selected climate action plans following a two-stage qualitative coding process. In stage 1, we defined a preliminary protocol of coding themes and categories according to common topics discussed in the literature[4,5,10,55,56]. These included the three dimensions of climate justice (distributive justice, procedural justice, and justice as recognition), as well as other key concepts related to justice and equity, mitigation sectors, and policy strategies. We define distributive justice as the fair allocation of the benefits and burdens of climate change and climate policy[10]; procedural justice refers to inclusive participation and engagement in decision-making processes[18,21]; and justice as recognition refers to the respect and valuing of all people in climate governance and requires the acknowledgement of historic and ongoing inequities as well as the pursuit of efforts to reconcile these inequities[9,10,18].

The preliminary protocol was pre-tested independently by each author on five climate action plans. This pre-testing enabled us to assess the robustness and clarity of the protocol and to refine coding categories before proceeding to the next stage. In stage 2, we began by using the preliminary protocol designed to code all plans within our sample. Here, we moved beyond a purely deductive coding approach and allowed new themes and categories to emerge and be redefined inductively from the data. All emerging categories were continuously discussed and agreed upon by both authors. As we adapted the protocol, we conducted iterative rounds of focused coding to homogenise our analysis across all plans. The final protocol included 98 subcategories and 18 main categories organized across six general themes: (1) distributional justice; (2) procedural justice; (3) justice as recognition; (4) justice in climate mitigation sectors; (5) key definitions; and (6) key sections where justice is articulated (Supplementary Table 2). We used NVivo 12 Pro software for all coding procedures.

Since our goal was to understand how cities are articulating climate justice with respect to climate mitigation, we only coded the sections and excerpts explicitly related to climate action in each plan. This means that in all plans not exclusively focused on mitigation, we did not code any chapters dedicated to climate adaptation, resilience, or any other sectors that were not explicitly recognised as a strategy to reduce greenhouse gas emissions. These restrictions helped us narrow our analysis to the climate justice discourses more directly associated with climate mitigation.

### Logistic regression analysis

We use ordinal logistic regression to identify local sociodemographic, economic, and political characteristics that may influence cities' level of engagement with justice in their climate action plans and to determine whether cities' attention to climate justice has increased over time.

Our dependent variable is an ordinal variable that measures cities' engagement with justice according to the city categories found through our analysis (i.e., Category 1: cities that do not articulate justice as a core feature of their climate plan; Category 2: cities that articulate justice as an aspiration; and Category 3: cities that are explicitly planning for justice). Our predictor variables are comprised of a set of cities' local sociodemographic, economic, and political factors. This data was obtained from the 2019 US Census estimates and the 2015–2019 American Community Survey. We also control for important characteristics of the climate plans themselves, including the year of publication, which was used to determine whether attention to justice has increased over time. All predictor variables were preselected through a literature review of previous research on urban climate action and climate justice[20,27,40,54,57]. Supplementary Table 5 presents the descriptive statistics and description of the dependent variable and all predictor variables considered for analysis.

The specification of our model was selected through forward and backward stepwise regression using the Akaike Information Criterion (AIC)[53]. We first fit a baseline model that included all predictor variables included in Supplementary Table 5 and applied the *stepAIC* function from R's *MASS* package with "both" as the direction of the selection technique. This command uses forward and backward

stepwise regression to select the model specification that minimizes the AIC. We also performed ANOVA to test whether the final model selected through *stepAIC* is better at capturing that data than the baseline model. This enabled us to verify that the additional variables present in the baseline model do not significantly improve the fit of the model. We tested the assumption of no multicollinearity through the Variance Inflation Factor (VIF) test, using the general rule that if the VIF for all parameters is less than 5, there is no evidence of multicollinearity. We also tested for the proportional odds assumption using the Brant test, which assesses whether the observed deviations from the ordinal logistic model are larger than what could be attributed to chance alone. Although there is no single agreed upon measure of goodness-of-fit for logistic regression[58], we decided to include the McFadden Pseudo-R$^2$ to assess the fit of our model[59]. Higher values of McFadden Pseudo-R$^2$ indicate a better model fit, and values between 20% and 40% are usually considered highly satisfactory.

### Reporting summary

Further information on research design is available in the Nature Research Reporting Summary linked to this article.

## Data availability

The data used and generated in this study (city climate action plans, qualitative content analysis, socio-demographic data, logistic regression analysis) have been deposited in an open repository: https://doi.org/10.5281/zenodo.7008298.

## Code availability

Codes used to produce this work are available in the open repository: https://doi.org/10.5281/zenodo.7008298.

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

## Acknowledgements

This work was supported by a Boston University Initiative on Cities Early Stage Urban Research Grant (ASG) and a National Science Foundation Research Traineeship (NRT) grant to Boston University (DGE 1735087, CVD). We thank Jessica Bajada Silva for her assistance with preliminary research.

## Author contributions

CVD: Writing—Original Draft, Methodology, Data Collection, Data Analysis; ASG: Writing—Review and Editing, Conceptualization, Methodology, Funding Acquisition, Project Supervision.

## Competing interests

The authors declare no competing interests.
