## [Peer Review File · Nature Communications]

U.S. cities increasingly integrate justice into climate planning and create policy tools for climate justiceReviewers' Comments:

Reviewer #1:

Remarks to the Author:

This is a very useful and well-written manuscript. Such reviews are very helpful to synthesize information from cities. They did a great job of categorizing the issues, visualizing graphics, and writing very simply. I recommend it's publication, with minor edits.

1) The intro noted that there's been very few studies looking at whether cities have addressed equity issues in mitigation, despite the many equity issues they raise. I'm not a mitigation expert, but can there really be none, except Hughes' research agenda paper? At the very least, it would be useful to elaborate on the issues raised in lines 47-48 - what are the equity and justice issues critics find concerning in these sectors? The authors could the better relate this to Figure 2b, to consider how well plans respond to the concerns identified by critical commentators.

2) There's repeated references to adaptation as a comparative case, but I wonder whether there are ways to better integrate the two, especially if the authors already have some familiarity with the adaptation space. Given the timing or what the literature says, do the authors think that the emphasis on equity and justice adaptation led climate mitigation plans to incorporate this (they often are developed by the same offices). Moreover, do the equity and justice aspects in the mitigation plans align well with the adaptation concerns and responses? For instance, are equity concerns now better integrated into mitigation plans than adaptation (see e.g. the Chu and Cannon or Broto and Bulkeley papers for juxtaposition?). Do they address similar or different sectors? This kind of discussion would support the authors' conclusion (starting line 225) about the challenges of integrating these concerns across sectors. That includes across the adaptation space, and whether these two issues are being considered in tandem or as wholly disparate issues.

Give this one more read for syntax. Eg. lines 70-71. The word fit is usually past tense, not "fitted". Table 1 should indicate the Y variable the other indicators are predicting.

Otherwise, great job!

Linda Shi

Reviewer #2:

Remarks to the Author:

Please find my comments in the attached reviewer report.

Reviewer #3:

Remarks to the Author:

The paper analyzes the climate plans of 58 U.S. cities to determine the extent to which they are incorporating justice and equity into their planning. The authors find that 40 cities do; half of these just mention equity and half have additional details for how they plan to meet their equity goals. The authors also identify four of the most common policy strategies used by the latter group of cities.

Overall, the paper does a good job of executing its goal of analyzing climate plans. My biggest concern about the paper is that it is not offering the field much that is new. Given the amount of previous research that has shown a huge implementation gap in city climate plans broadly, it is not clear what we should even make of cities that include justice in their plans, and the authors do not provide an answer to this either. It is also not clear to me that cities that are planning for justice (rather than

aspiring to justice) are better positioned for implementation success. It will be the responsibility of a climate or sustainability office to turn any plan's aspirations into reality anyway, and having an equity advisory board mentioned in the plan will be what makes the difference. If the two main conclusions are that (a) justice and equity are growing but not central foci in urban climate plans, and that (b) plans lack specificity on implementation, I do not know that there is much new here for the field.

The authors highlight many of what to me are the important issues for the field in their Discussion: that nearly half of the cities don't have plans; that really achieving what is called a just urban transition is intersectional (as opposed to their sectoral analysis); and cities need to develop strategies for operationalizing these goals. The authors do not evaluate their findings in any real way (are justice indicators going to get cities where they want to go?) or provide policy recommendations.

I was surprised when reviewing the tables and figures in the back that the authors did not say more about the results of their regression analyses. There could be more to draw from there in terms of understanding the "types" of cities that are pivoting toward justice (at least on paper).

Again, I think the study is well-executed but am not seeing how it is an "important advance of significance to specialists." My critique is not solely that the study is descriptive, but that the descriptive information provided may not be providing innovative insights given the precarious nature of urban climate plans.

RESPONSE TO REVIEWERS

We are grateful for the thoughtful and helpful feedback from the three reviewers. We are glad all three reviewers found the manuscript clear and sound. In our revision, we endeavored to address the reviewers' comments and suggestions while maintaining the clear writing and flow of the manuscript. We believe the revisions have improved the manuscript and made the contribution even stronger.

Detailed responses to the reviewer comments are provided below. All changes referred to in these responses are also highlighted in yellow in the manuscript.

REVIEWER 1

Reviewer's comment	Authors' response
This is a very useful and well-written manuscript. Such reviews are very helpful to synthesize information from cities. They did a great job of categorizing the issues, visualizing graphics, and writing very simply. I recommend it's publication, with minor edits. 1) The intro noted that there's been very few studies looking at whether cities have addressed equity issues in mitigation, despite the many equity issues they raise. I'm not a mitigation expert, but can there really be none, except Hughes' research agenda paper? At the very least, it would be useful to elaborate on the issues raised in lines 47-48 - what are the equity and justice issues critics find concerning in these sectors? The authors could the better relate this to Figure 2b, to consider how well plans respond to the concerns identified by critical commentators.	We edited the introduction to better explain our contributions. In particular, we highlight that the main research gap we are addressing lies in the small number of studies that have directly analyzed the integration of justice concerns in urban climate mitigation planning. The reviewer is correct in that there are several case studies looking at the justice or equity aspects of specific policies (e.g., Bulkeley et al., 2014). However, we still lack comprehensive studies looking at how cities formally articulate and include justice in climate mitigation plans across a large sample of cities. Our revision to lines 44 - 60 of the introduction section clarifies this knowledge gap and why it matters. We were not entirely sure how to interpret the comment about linking the concerns to Figure 2b. Nonetheless, we hope we addressed the Reviewer's concerns by more directly articulating (a) the lack of comprehensive knowledge across cities (revised Intro) and (b) adding some discussion of the sector analysis (revised lines 317 - 326 in section "Moving toward just urban transitions").
2) There's repeated references to adaptation as a comparative case, but I wonder whether there are ways to better integrate the two, especially if the authors already have some familiarity with the adaptation space. Given the timing or what the literature says, do the authors think that the emphasis on equity and justice adaptation led climate mitigation plans to incorporate this (they often are developed by the same offices).	We edited and added text throughout the manuscript to make it clear when we are explicitly comparing results between our analysis and previous studies focused on climate adaptation plans. See for example:  - Section "Engagement with justice in urban climate action plans", lines 113 – 133.

Moreover, do the equity and justice aspects in the mitigation plans align well with the adaptation concerns and responses? For instance, are equity concerns now better integrated into mitigation plans than adaptation (see e.g. the Chu and Cannon or Broto and Bulkeley papers for juxtaposition?). Do they address similar or different sectors? This kind of discussion would support the authors' conclusion (starting line 225) about the challenges of integrating these concerns across sectors. That includes across the adaptation space, and whether these two issues are being considered in tandem or as wholly disparate issues.	 - Section “Articulations of justice and equity”, lines 149 – 158 and 161 – 169. - Section “Operationalising just climate policies”, lines 217 – 219. - Section “Moving toward just urban transitions”, lines 317 – 326.
Give this one more read for syntax. Eg. lines 70-71. The word fit is usually past tense, not "fitted". Table 1 should indicate the Y variable the other indicators are predicting. Otherwise, great job! Linda Shi	Following the formatting used in other Nature family papers, we edited the title of Table 1 and added a description that explicitly defines our dependent variable (Table 1 is now located between lines 135 – 140). We conducted a thorough syntax review and made minor edits accordingly.

REVIEWER 2

Reviewer’s comment	Authors’ response
This is a well-written piece that engages with climate justice in cities, which is a topic of growing interest among urban climate researchers. Will the work be of significance to the field and related fields? How does it compare to the established literature? If the work is not original, please provide relevant references/ What are the noteworthy results? The significance and major contribution of this study is to examine manifestations of climate justice in a large set of cities in the US. I agree with the premise set out by the authors, which is that most analyses of urban climate justice have focused on single or few case studies. Thus, the paper is clearly making an empirical contribution. Having said that, this is not the first study documenting climate justice by comparing patterns across cities. Bulkely et al (2013, already cited in the paper), conducted a comparative study of aspects of justice addressed in a large set of climate interventions in cities nearly a decade ago. There are also studies of a similar	To respond to this comment, as well as comments from the other two reviewers, we edited the introduction to better highlight our contributions and clarify the research gap we are addressing through our study. Lines 61 - 71 of the introduction section now explicitly spells out the contributions by articulating the following new findings and analyses:  - Large cities across the U.S. are increasingly incorporating justice into their climate action plans, particularly in the last five years. - The recognition of historical patterns of racial segregation, disinvestment, environmental injustice, and exclusion is becoming more common in recent plans. - Attention to justice is not equally distributed across mitigation sectors.

nature, such as examination of equity and justice dimensions in urban sustainability plans, including (Hess et al, 2021 and Schrock et al, 2015, already cited in this paper), or others such as:  - Castán Broto, V., & Westman, L. (2017). Just sustainabilities and local action: Evidence from 400 flagship initiatives. Local Environment, 22(5), 635-650. - Pearsall, H., & Pierce, J. (2010). Urban sustainability and environmental justice: Evaluating the linkages in public planning/policy discourse. Local Environment, 15(6), 569-580. - Wolch, J. R., Byrne, J., & Newell, J. P. (2014). Urban green space, public health, and environmental justice: The challenge of making cities „just green enough“. Landscape and urban planning, 125, 234-244. Existing studies have also focused specifically on large cities in the US. As these previous contributions have mapped specific trends in justice in urban planning, a key question is what new results have emerged from this new analysis of climate plans in the US context. My understanding is that the authors have documented a growing engagement in justice, compared to the studies listed above and as shown by their own analysis of recently adopted plans, which is a noteworthy result. In terms of the content of the plans, the results appear to follow previous findings, in terms of showing a greater emphasis on equity and distribution concerns rather than structural conditions; however, the focus in these plans on “racial segregation, disinvestment, environmental injustice, and exclusion” is an interesting and relatively novel result. The discovery of concrete tools to integrate justice considers is also valuable.	 - We uncover four concrete policy tools cities are using to implement and evaluate work toward “just urban transitions”. - We analyse local factors that may influence cities’ level of engagement with justice into their climate action plans. These contributions are aligned to the findings that Reviewer 2 highlighted as valuable empirical and theoretical contributions.
In terms of theoretical contributions, the conclusions that are discussed in the section called Moving toward just urban transitions seem to mainly reflect insights that are relatedly well established in this field. The authors conclude that: there is a growing interest in justice, but that urban policymakers need to focus more on these issues; that justice planning needs to move beyond sectoral thinking; and that the capacity of planning tools to tackle structural conditions of justice is unknown. While I agree with these points, I also think that the study contains several findings that could be discussed in further depth to advance the contributions of the study further. For instance:	We too are intrigued by “why” engagement varies across different sectors. In the revised manuscript, we add text throughout the results section that speaks to some of the questions and trends highlighted by Reviewer 2 and highlight the space for new research. Specifically, we made the following changes:  - We have added a new paragraph at the end of the section “Engagement with justice in urban climate action plans” (see lines 113 – 133). The new text provides a complete description of the results from our regression analysis and reflections on

 - 42% of cities have not considered justice aspects at all. Is there any trends in terms of characteristics of cities that do adopt cities (income level, politics at higher levels of government, geography) that suggests there is a pattern to which cities do engage with justice (and perhaps even which forms of justice they engage with)? I was surprised to see that the authors have conducted a rather complex regression analysis of such factors (supplementary Table 1), but not discussed these results in the text? - Figure 2 is fascinating and I imagine that a deeper exploration of trends could lead to further insights. For example, why does engagement with justice vary across sectors? Can the authors link this with strategies of certain social movements, or history of engagement in particular policy issues? Why have justice considerations failed to appear in relation to EVs, water, and waste and what problems are overlooked in these cases? Likewise, it is interesting that what is often labelled „justice in recognition“ (frequently overlooked in both policy and research) here can be understood more clearly in the data on engagement with structural inequalities based on race, income, gender, and disability. I would encourage the authors to reflect on why plans have been able to address racism or income disparities but not gender and ability? 	our findings. The regression analysis identifies trends between cities that do not articulate justice, cities that articulate justice as an aspiration, and cities explicitly planning for justice.* We also make a call for future research that is needed to better understand the dynamics behind the trends we found.  - We have added reflections on the limited attention that gender and disability have received across cities (See Section “Articulations of justice and equity”, lines 161 - 189). - We have added insights on the types of narratives we can expect in the aftermath of the COVID-19 pandemic. (See Section “Articulations of justice and equity”, lines 170 - 177). - We have added a comparison of attention justice across sectors in climate mitigation and adaptation (See “Moving toward just urban transitions”, lines 318 - 326). * We want to clarify that the regression analysis focuses on the 58 cities with climate action plans. (The 42 cities that have not developed climate action plans are not included in the analysis). While it may be interesting to examine differences in cities with and without climate action plans, we elected to keep the focus tightly on justice so did not include this additional analysis.
Is there enough detail provided in the methods for the work to be reproduced? Overall, the authors present a detailed account of their methodology. However, I could not find much explanation for the details of the coding. The authors refer to three widely used principles of justice (distribution, procedure, recognition), but have not discussed definitions of these, nor explained how they operationalized these concepts. The main text refers to a supplementary Table 2 as containing their coding guide, but the table in the annex seems to contain an overview of equity advisory boards. This aspect of their methods seems very important to me, as justice dimensions of urban plans rarely are explicit but need to be captured through indirect criteria. Results will depend greatly on how the authors	Supplementary Table 2 includes all the coding categories or criteria that were used to identify the different articulations of the three dimensions of justice. Supplementary Table 2 can be found in the document “Supplementary Tables”. We have also added a brief description of the definitions we considered for each of the three dimensions of justice (see lines 412 - 419 of the sub-section “Coding protocols and procedures” in the Methods section). Please note: Supplementary Table 2 is distinct from the tables in the main text of the manuscript. The table with the overview of equity advisory boards is in the main text – it was labeled Table 2 in the original submission and has been renumbered as Table 3 in this revision.

have interpreted the principles and captured them in their sample of policy documents.	
Minor comments Does the work support the conclusions and claims, or is additional evidence needed? Yes, the evidence clearly supports the conclusions and claims. Are there any flaws in the data analysis, interpretation and conclusions? Do these prohibit publication or require revision? The data analysis is sound and convincing. Is the methodology sound? Does the work meet the expected standards in your field? The methodology follows established processes of policy document reviews in this field of research. One aspect of the methodology that I reacted to was the selection of large cities for analysis. My understanding is that large urban areas are the comparatively well-documented in sustainability and climate planning research – why focus on these cities and not smaller municipalities? Whether this is because there is no data on smaller cities or if the selection was based on another logic, the decision should be specified.	We have added our rationale for focusing on large cities in the first paragraph of the Methods section (see lines 379 – 385).

REVIEWER 3

Reviewer's comment	Authors' response
The paper analyzes the climate plans of 58 U.S. cities to determine the extent to which they are incorporating justice and equity into their planning. The authors find that 40 cities do; half of these just mention equity and half have additional details for how they plan to meet their equity goals. The authors also identify four of the most common policy strategies used by the latter group of cities. Overall, the paper does a good job of executing its goal of analyzing climate plans. My biggest concern about the paper is that it is not offering the field much that is new. Given the amount of	We have edited and added new text in several parts of the paper to highlight the contributions of our research and explicitly spell out why understanding how cities articulate and integrate justice concerns into formal climate action planning documents is important for academics and practitioners in the field of urban climate governance and politics. In lines 44 - 60 of the Introduction, we highlight that the main research gap we are addressing lies in the small number of studies that have directly analyzed the integration of justice concerns in urban climate mitigation planning. While it is true

previous research that has shown a huge implementation gap in city climate plans broadly, it is not clear what we should even make of cities that include justice in their plans, and the authors do not provide an answer to this either.

It is also not clear to me that cities that are planning for justice (rather than aspiring to justice) are better positioned for implementation success. It will be the responsibility of a climate or sustainability office to turn any plan's aspirations into reality anyway, and having an equity advisory board mentioned in the plan will be what makes the difference.

If the two main conclusions are that (a) justice and equity are growing but not central foci in urban climate plans, and that (b) plans lack specificity on implementation, I do not know that there is much new here for the field.

The authors highlight many of what to me are the important issues for the field in their Discussion: that nearly half of the cities don't have plans; that really achieving what is called a just urban transition is intersectional (as opposed to their sectoral analysis); and cities need to develop strategies for operationalizing these goals. The authors do not evaluate their findings in any real way (are justice indicators going to get cities where they want to go?) or provide policy recommendations.

that implementation gaps have consistently been identified in urban climate plans, analyzing these formal planning documents is still important to understand how normative goals such as justice and equity are institutionalized in city governments and to provide benchmarks for new cities to integrate justice concerns into their climate efforts.

In response to this comment, as well as the suggestions from the other two reviewers, we also edited the introduction to highlight our contributions. Lines 61 – 71 of the introduction section now explicitly spell out the following empirical and theoretical contributions:

- Large cities across the U.S. are increasingly incorporating justice into their climate action plans, particularly since the last five years.
- The recognition of historical patterns of racial segregation, disinvestment, environmental injustice, and exclusion is becoming more common in recent plans.
- Attention to justice is not equally distributed across mitigation sectors.
- We uncover four concrete policy tools cities are using to implement and evaluate work toward “just urban transitions”.
- We analyse local factors that may influence cities’ level of engagement with justice into their climate action plans.

We also highlight again the importance of having urban climate plans that incorporate justice in lines 300 - 311 of the section “Moving toward just urban transitions”. This directly address Reviewer’s 3 concern of why climate plans are still relevant tools even if scholars have identified implementation gaps.

Reviewer 3 correctly points out that, in the end, it is implementation and not just the development of climate plans that will make the difference for climate justice. We share this opinion and highlight the current lack of implementation guidance in most cities’ climate plans (see section Moving toward just urban transitions”, lines 339 - 344). In lines 344 - 354, we also point out that the pioneer cities we identified and the four policy tools we uncovered in our analysis can serve as a benchmark for other cities, and we explain that

	evaluating the “success” of these implementation tools is not possible through our analysis of climate action plans, particularly because these tools and plans have been developed only in the past few years. While this is a limitation of our analysis, we believe our findings provide helpful data for future case study analyses focused on implementation. We suggest case studies of pioneering cities as important next steps for the research community.
I was surprised when reviewing the tables and figures in the back that the authors did not say more about the results of their regression analyses. There could be more to draw from there in terms of understanding the "types" of cities that are pivoting toward justice (at least on paper).	In response to this comment, as well as Reviewer 2’s suggestions, we have added additional text highlighting the results from our logistic regression analysis. We have added a new paragraph at the end of the section “Engagement with justice in urban climate action plans” (see lines 113 – 133). Here, we provide a thorough description of the results from our regression analysis and reflect on our findings. The regression analysis identifies trends between cities that do not articulate justice, cities that articulate justice as an aspiration, and cities explicitly planning for justice. We also make a call for future research that is needed to better understand the dynamics behind the trends we found (See lines 132 - 133).
Again, I think the study is well-executed but am not seeing how it is an "important advance of significance to specialists." My critique is not solely that the study is descriptive, but that the descriptive information provided may not be providing innovative insights given the precarious nature of urban climate plans.	See comments above.

Reviewers' Comments:

Reviewer #1:

Remarks to the Author:

The authors have addressed my previous concerns. I recommend the manuscript for publication.

Reviewer #2:

Remarks to the Author:

Thank you for sending me the revised version of this manuscript. The authors have done a thorough job responding to my comments, as well as the comments from the other two reviewers. In particular, I find that the paper is significantly strengthened through the efforts of the authors to: 1. situate the manuscript more clearly in relation to a gap in previous research; 2. unpack the results further by providing more information on the regression analysis and additional detail on the articulations of justice in current plans; and 3. identify directions for future research.

Through these revisions, all major concerns that I raised in my original set of comments have been addressed. My opinion is that the quality of the manuscript now matches the requirement of Nature Communications and I would recommend its publication.

Reviewer #3:

Remarks to the Author:

The authors have done an adequate job of highlighting what they see as their key contributions, and generally addressing concerns raised by reviewers.

RESPONSE TO REVIEWERS

REVIEWER'S COMMENTS

Reviewer #1 (Remarks to the Author):

The authors have addressed my previous concerns. They recommend the manuscript for publication.

Reviewer #2 (Remarks to the Author):

Thank you for sending me the revised version of this manuscript. The authors have done a thorough job responding to my comments, as well as the comments from the other two reviewers. In particular, I find that the paper is significantly strengthened through the efforts of the authors to: 1. situate the manuscript more clearly in relation to a gap in previous research; 2. unpack the results further by providing more information on the regression analysis and additional detail on the articulations of justice in current plans; and 3. identify directions for future research.

Through these revisions, all major concerns that I raised in my original set of comments have been addressed. My opinion is that the quality of the manuscript now matches the requirement of Nature Communications and I would recommend its publication.

Reviewer #3 (Remarks to the Author):

The authors have done an adequate job of highlighting what they see as their key contributions, and generally addressing concerns raised by reviewers.

RESPONSE TO REVIEWERS

We are grateful for the thoughtful and helpful feedback we received from the three reviewers in previous submissions of the manuscript. We believe the reviewers' feedback significantly improved the manuscript and we are glad all the reviewers are satisfied with the latest revisions.